# LLM Data Selection and Utilization via Dynamic Bi-level Optimization

**Yang Yu** [1 2 3]   **Kai Han** [3]   **Hang Zhou** [3 4]   **Yehui Tang** [3]   **Kaiqi Huang** [1 2]   **Yunhe Wang** [3]   **Dacheng Tao** [5]

## Abstract

While large-scale training data is fundamental for developing capable large language models (LLMs), strategically selecting high-quality data has emerged as a critical approach to enhance training efficiency and reduce computational costs. Current data selection methodologies predominantly rely on static, training-agnostic criteria, failing to account for the dynamic model training and data interactions. In this paper, we propose a new Data Weighting Model (DWM) to adjust the weight of selected data within each batch to achieve a dynamic data utilization during LLM training. Specially, to better capture the dynamic data preference of the trained model, a bi-level optimization framework is implemented to update the weighting model. Our experiments demonstrate that DWM enhances the performance of models trained with randomly-selected data, and the learned weighting model can be transferred to enhance other data selection methods and models of different sizes. Moreover, we further analyze how a model's data preferences evolve throughout training, providing new insights into the data preference of the model during training.

## 1. Introduction

The success of modern language models has demonstrated the critical role that large-scale pre-training data plays in shaping their performance (Brown et al., 2020; Touvron et al., 2023a). The diversity and scale of the training data are essential for enabling the model to generalize across various tasks and domains (Bai et al., 2024; Xia et al., 2024; Zhou et al., 2024). As the scale of the pre-training data increases, language models exhibit a remarkable capacity to perform downstream tasks with minimal task-specific tuning, showcasing the power of data-driven approaches in natural language processing (Kaplan et al., 2020).

Though the power of large language models rises with the use of enormous and ever-growing datasets for pre-training, naively training a model on all available data may not be optimal, as the quality of available data varies, and the training increases the carbon footprint and financial costs. Recently, there is increasing evidence that choosing the right training data is more essential for producing state-of-the-art large language models (Albalak et al., 2024; Zhao et al., 2023), and researchers have focused on studying data selection, the mechanism to determine which candidate data to include in the training, to improve the training efficiency of model pre-training. Specially, these data selection methods usually utilize referring data or referring models for an effective selection process. For example, in DSIR (Xie et al., 2023), Wikipedia and books are used as the high-quality data to help data classification. In DsDM (Engstrom et al., 2024) and MATES (Yu et al., 2024), LAMBADA dataset (Paperno et al., 2016) is used as the target dataset to help evaluate the influence of the candidate data, *i.e.*, the decrease of the loss in the target dataset when the model is trained with and without the candidate data. Besides, referring models are also utilized to help evaluate the quality of the data. The relationship between the perplexity of reference models (e.g., Llama) applied to candidate datasets and data quality is examined, with perplexity proposed as a potential metric for assessing data quality (Ankner et al., 2024). And QuRating (Wettig et al., 2024) utilizes the data preference of GPT-3.5-turbo to train the data rating model.

Though these methods filter out data and decrease the training cost, they merely focus on the data selection isolatedly without considering the dynamic training process of LLMs. On the one hand, most of the previous methods selected data before model training, ignoring the dynamic data preference of the model during training. On the other hand, the data samples in existing methods are selected separately and utilized within a batch indiscriminately, without considering the joint effects between different samples. In fact, the data

---

[1]School of Artificial Intelligence, University of Chinese Academy of Sciences [2]The Key Laboratory of Cognition and Decision Intelligence for Complex Systems, Institute of Automation, Chinese Academy of Sciences [3]Huawei Noah's Ark Lab [4]College of Intelligence and Computing, Tianjin University [5]Nanyang Technological University. Correspondence to: Kaiqi Huang <kqhuang@nlpr.ia.ac.cn>, Yunhe Wang <yunhe.wang@huawei.com>.

*Proceedings of the $42^{nd}$ International Conference on Machine Learning*, Vancouver, Canada. PMLR 267, 2025. Copyright 2025 by the author(s).

samples interact with each other, and the combination of them determines the model update direction together. Hence, the existing methods which select data separately and ignore the data utilization will limit the potential performance of the trained LLMs with selected data.

In this paper, to improve the data utilization for large language models training with selected data, we propose a bi-level optimization framework to capture the dynamic data preference of the model, and the joint effects of different data samples. In the framework, a plug-and-play Data Weighting Model (DWM) is introduced to weigh the data samples within each batch during model training, and therefore focuses on the joint effects of selected data. Specifically, to guarantee the weighting model knows the data preference of the trained model, we introduce a bi-level optimization to help learn the weighting model. The lower level will first optimized the trained model with data weighted by the weighting model, and the upper level will optimized the trained model updated by the lower-lever optimization, where the weighting model can be optimized with the help of the chain rule. Furthermore, to better capture the dynamic data preference of the trained model, we learn the DWM via the above bi-level optimization at different stages during model training, and hence learn the data preference dynamically and adaptively.

We conduct extensive experiments to validate the effectiveness of DWM. First, using randomly-selected data from SlimPajama, we pre-train a 370M model from scratch. The model trained with DWM and randomly selected data outperforms both models trained with randomly-selected data and those with carefully selected data. We then transfer DWM to a larger LLM (*i.e.*, 1.3B) and other data selection methods, which also achieve a consistent performance improvement. Finally, we further analyze how the weighting model preferences evolve during training to provide more insights about the model data preference.

## 2. Related Work

We review existing data selection methods for large language models and analyze their relevance to our proposed approach.

**Categories of Data Selection Methods**   As discussed in RegMix (Liu et al., 2024), existing data selection methods for LLM pre-training can be categorized into three categories: (1) Token-level selection deals with the filtering of tokens, like Rho-1(Lin et al., 2024); (2) Group-level selection groups data into pools and focuses on the pool-mixing, like RegMix (Liu et al., 2024); (3) Sample-level selection is about choosing individual training examples. The category of sample-level selection methods can be further divided into two types: referring data-based methods and referring

model-based methods. Our Method also belongs to sample-level data selection.

**Sample-Level Data Selection**   Sample-level data selection methods can be classified into two sub-types: reference data-based and reference model-based approaches. Referring data-based methods mainly select data samples with the help of other data. For example, in DSIR (Xie et al., 2023), Wikipedia and books are utilized as high-quality reference data to facilitate data classification. In DsDM (Engstrom et al., 2024) and MATES (Yu et al., 2024), the LAMBADA dataset (Paperno et al., 2016) serves as the target dataset to assess the impact of candidate data—specifically, by measuring the reduction in loss on the target dataset when the model is trained with versus without the candidate data. Additionally, a line of research investigates leveraging reference models to guide data selection, *i.e.*, referring model-based methods. For instance, the relationship between the perplexity of reference models (e.g., LLaMA) on candidate datasets and data quality has been analyzed, with perplexity proposed as an effective metric (Ankner et al., 2024). Another approach, QuRating (Wettig et al., 2024), leverages the implicit data preferences of the referring model GPT-3.5-turbo to train a model for data rating. However, most data selection methods remain isolated and fail to consider the dynamic training process of the model. While MATE partially accounts for the multi-stage training process of the model, it still overlooks the joint effects of selected data, as the chosen data is used indiscriminately during training. In contrast, our method focuses more on data utilization and proposed DWM to capture the dynamic data preference of the model, and the joint effects of different data samples during model training.

## 3. Approach

### 3.1. Motivation

The performance of large language models is strongly influenced by the scale and diversity of pre-training data, and data selection mitigates the inefficiency of model pre-training caused by the variations of data quality and data redundancy. Despite these advances, existing methods typically focus on pre-training data selection without considering the nature of model training. Most approaches select data in isolation before training and use data indiscriminately during training, overlooking the interactions between data points and the shifty data preference of the trained model.

In this paper, to better utilize selected data for model pre-training, and capture the nature of model training, we propose a novel plug-and-play Data Weighting Model (DWM) to assist model training. DWM will be used to weigh the data in a batch, and hence adapt to the joint interaction

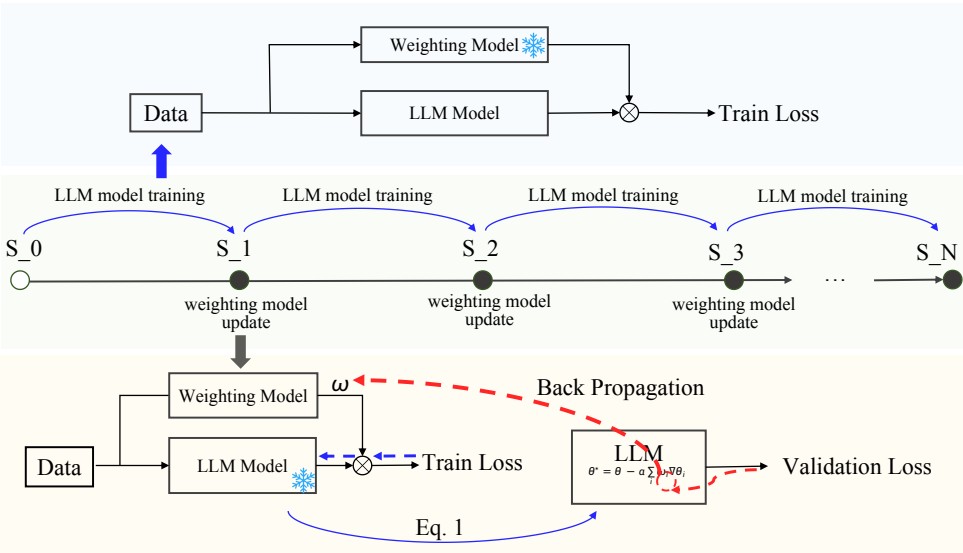

*Figure 1.* The framework of the proposed bi-level optimization process with DWM, where the LLM model and the weighting model are trained alternated. During model training, the weighting model is frozen and the pre-training loss is the weighted sum of the loss of each data sample in one batch. Besides, to capture the data preference of the trained model, the weighting model is updated to minimize the validation loss of the model trained with the weighting model with the chain rule.

between data during training. And in order to capture the dynamic data preference of the trained model, a bi-level optimization framework is introduced to help update the weighting model by stages. The framework of DWM is shown in Figure 1, where the model training and the weighting model update are alternated by stages. In the following, we will present the bi-level optimization process and the details of the update of the trained LLM model and the weighting model.

### 3.2. Dynamic Bi-level Optimization

To jointly optimize both the LLM and the importance of the training data (i.e., DWM), while accounting for the dynamic nature of LLM training, we propose a dynamic bi-level optimization framework for iterative improvement. This framework involves updating both the trained model and the weighting model to capture the evolving data preferences. The weighting model learns the data preferences of the current LLM based on the interactions within a single batch and provides better update directions by assigning different weights to the data samples. The challenge lies in effectively optimizing this weighting model. A straightforward approach is to evaluate the weighting model using the standard auto-regressive loss on the training data. However, since the pre-training loss is also dependent on the output of the weighting model, updating the weighting model with this loss may lead to sub-optimal solutions—such as setting the weight of all data or easily predicted data points to zero in an attempt to minimize training loss. This, however,

does not contribute to the performance improvement of the trained model.

To achieve a better optimization of the weighting model and help the training of the trained model, in this work we propose a novel weighting model evaluation metric called *weight influence*, which is defined as:

> *The performance of the trained LLM model in the validation dataset when optimized with the weighting model.*

Then we can optimize the weighting model by maximizing its weight influence. Concretely, the process of jointly considering the trained model $\theta$ and the weighting model $\theta_w$ for updating can be viewed as a bi-level optimization problem:

$$
\begin{aligned}
\max_{\theta_w} \quad & R_{\text{val}}(\theta^*(\theta_w)) \\
\text{s.t.} \quad & \theta^*(\theta_w) = \arg\min_{\theta} L_{\text{train}}(\theta, \theta_w),
\end{aligned}
\tag{1}
$$

where the first term is to optimize $\theta_w$ by maximising the reward performance $R_{\text{val}}$ (e.g., accuracy) on validation set, while the second term is the LLM training based on training data re-weighted by the data weight model. And therefore, when optimizing the upper level object, *i.e.*, the performance of the trained LLM model in the validation set, the weighting model can be optimized with the help of the chain rule. Via this bi-level optimization, the weighting model learns how to weigh sampled data to improve the generalization of the trained model, which is of benefit to the model pre-training.

In the following, we'll describe the details of the bi-level optimization of each part.

### 3.3. LLM Training

In DWM, model pre-training is similar to the normal pre-training process except for the utilization of the weighting model, where data in one batch will be weighted instead of used indiscriminately. The contribution weight of each sample $X_i$ to the training of LLM can be obtained by the introduced DWM:

$$\omega_i = \theta_w(X_1, X_2, \cdots, X_{bs})_i, \tag{2}$$

where $i = 1, 2, \cdots, bs$ is the sample index and $bs$ is the total number of data samples in one batch. Here DWM will consider the data interaction in one batch to weigh the contribution of each sample.

Here we use the output of the weighting model $\theta_\omega$ to weigh the training loss of each data sample, and therefore, the loss during pre-training stage is:

$$L_{train}(\theta, \theta_\omega) = \sum_i^{bs} \omega_i L_{train,i}(\theta). \tag{3}$$

The training loss $L_{train,i}$ is the normal auto-regressive loss for LLM training. It is important to note that DWM remains fixed during LLM training for improving data utilization and ensuring stable learning dynamics, allowing the model to focus on optimizing the LLM parameters without interference from fluctuating data weights.

### 3.4. Data Weighting Model Training

As discussed above, the weighting model is optimized to maximize the trained model performance in the validation dataset. Similar to the motivation behind meta-learning, the goal of learning the weighting model is to capture a more effective data weighting mechanism, thereby improving the generalization of the trained model in later stages of training.

Referring to Eq. 1, it is worth noting that the trained model is updated with the output of the weighting model, and the updated model is evaluated in the validation set. Therefore we could optimize the weighting model directly by the trained model performance in the validation dataset with the chain rule. Concretely, we will first record the updated trained model parameters explicitly, which is:

$$\begin{aligned} \theta^* &= \theta - \alpha \sum_i^{bs} \omega_i \nabla \theta_i \\ &= \theta - \alpha \sum_i^{bs} \omega_i \frac{\partial L_{train,i}(\theta)}{\partial \theta}, \end{aligned} \tag{4}$$

where $\alpha$ is the learning rate of the model and $\nabla \theta_i$ is the gradient by training the model with data sample $X_i$. Then

the updated trained model will be used to calculate the validation performance:

$$\begin{aligned} R_{val}(\theta^*) &= \sum_i^M R_{val,i}(\theta^*) \\ &= \sum_i^M R_{val,i}(\theta - \alpha \sum_i^{bs} \omega_i \nabla \theta_i), \end{aligned} \tag{5}$$

where $M$ is the number of data samples in the validation set.

Consequently, the weighting model can be updated by optimizing the validation performance of the trained model in chained:

$$\begin{aligned} \frac{\partial R_{val}(\theta^*)}{\partial \theta_w} &= \frac{\partial R_{val}(\theta^*)}{\partial \theta^*} \frac{\partial \theta^*}{\partial \theta_w} \\ &= \sum_i^M \frac{\partial R_{val,i}(\theta^*)}{\partial \theta^*} \cdot (-\alpha \sum_j^{bs} \frac{\partial \omega_j}{\partial \theta_\omega} \nabla \theta_j) \\ &= -\alpha \sum_i^M \frac{\partial R_{val,i}(\theta^*)}{\partial \theta^*} \cdot \sum_j^{bs} \frac{\partial \omega_j}{\partial \theta_\omega} \nabla \theta_j. \end{aligned} \tag{6}$$

Hence the weighting model can be optimized to improve the performance of the trained model. The LLM is kept fixed while updating DWM to prevent any potential leakage of validation data.

### 3.5. Multi-stage Alternative Iteration

As discussed above, we update the trained model or the weighting model and keep the other fixed to ensure the stability of training or avoid the knowledge leakage. In order to capture the dynamic data preference of the trained model, here we employ a multi-stage alternative iteration process to jointly optimize the trained model and the weighting model. In this iteration process, the trained model parameters $\theta$ and weighting model parameters $\theta_w$ are updated in a stage-wise manner. Concretely, starting from parameters $\theta^{t-1}$ and $\theta_w^{t-1}$ inherited from stage $t-1$, the iteration proceeds at stage $t$ as follows:

1. Weighting Model Update. Fixing the trained model, we first update $\theta_w$ referring to Eq. 6:

$$\theta_w^t = \theta_w^{t-1} + \eta \nabla_{\theta_w} R_{val}\left(\theta^{t-1,*}(\theta_w)\right). \tag{7}$$

2. Trained Model Update. With the updated weighting model $\theta_w^t$, we then optimize the trained model by minimizing the training loss referring to Eq. 3 in this stage:

$$\theta^t = \arg\min_\theta L_{train}(\theta, \theta_w^t). \tag{8}$$

Each stage $t \in \{1, 2, \cdots, T\}$ strictly enforces an alternating update order to resolve the interdependence between $\theta$ and

$\theta_w$. This iterative process ensures that both models mutually enhance each other's performance through progressive refinement.

In summary, the proposed method effectively optimizes data utilization during training by dynamically adjusting data weights through DWM. By capturing and leveraging the model's evolving data preferences, DWM improves both the training effectiveness and generalization of the model. This approach allows for better utilization of selected data, leading to enhanced performance even with randomly selected data. Additionally, the trained weighting model can be transferred to larger models or applied to other data selection strategies, making it a robust solution for optimizing LLM training across various scenarios.

## 4. Experiment Setup

**Implementation Details** To evaluate different data selection methods, we utilize the training data selected from the popular dataset SlimPajama (Soboleva et al., 2023), which is the largest multi-corpora, open-source dataset for training large language models. SlimPajama is a cleaned and deduplicated version of the RedPajama dataset (Weber et al., 2024), includes 627 billion tokens and provides the metadata about the domain information for each sample. To verify the performance of different data selection methods for model pre-training, we adopt the model architecture from Llama-2 (Touvron et al., 2023b) at the scale of 370 million and 1.3 billion parameters separately to do the model pre-training. Following the principles of the scaling law and the QuRating (Wettig et al., 2024) framework, a total of 30 billion tokens are selected by different methods to train the model. As for the weighting model, we adopt the architecture from Llama-2 with 370 million parameters to learn the sample embedding, with additionally one attention block and two linear layers to generate the data weights considering the combination of the data in a batch. We set the sequence length to 1024 and the batch size to 4096, with a micro batch size ($bs$) as 8 to balance the effectiveness of the weighting model and the GPU memory constraints. Following DSDM and MATES, we adopt LAMBADA (Paperno et al., 2016) as the validation set, which is a widely-used language modeling task and often serves as a validation task for language model pre-training, and split the training process into 5 stages to learn the dynamic data preference. At the first stage, the trained model learns from scratch and weighs sampled data uniformly, and in the second stage, the weighting model will be partially initialized from the previous trained model and be continued trained in the later stage. More details can be found in Appendix A.

**Baselines** It is worth noting that our method DWM aims to improve the training data utilization of model pre-training,

and is orthogonal to existing data selection methods. In this paper, to verify the effect of DWM, we first compare DWM using random-selected data to naive random selection, as well as state-of-the-art data selection baselines including DSIR (Xie et al., 2023) and QuRating (Wettig et al., 2024). Here DSIR uses Wikipedia and books as target data distribution and selects data classified in the target distribution. QuRating first uses the model GPT-3.5-turbo (OpenAI, 2023) as the referring model to judge the data samples in different dimensions, then uses the model's preference data to train data quality rating models to select data. Moreover, we also provide results of DWM transferred to the data selected by DSIR or QuRating to justify its scalability.

**Evaluation Benchmarks** To evaluate the effectiveness of DWM, we compared the performance of the pre-trained models trained with DWM and different data selection baselines on the widely-used lm-evaluation-harness framework [1]. Following Wettig et al. (2024) and Xie et al. (2023), we perform a holistic evaluation of pre-trained models across 9 downstream tasks (ARC-easy/challenge (Clark et al., 2018), SciQA (Auer et al., 2023), LogiQA (Liu et al., 2020), BoolQ (Clark et al., 2019), OBQA (Mihaylov et al., 2018), HellaSwag (Zellers et al., 2019), PIQA (Bisk et al., 2020), and WinoGrande (Sakaguchi et al., 2021)), comprising reading comprehension tasks, commonsense reasoning and knowledge question answering. To assess both inherent knowledge and in-context learning capabilities, we evaluate the models under zero-shot and two-shot settings referring to MATES (Yu et al., 2024). For each evaluation task, normalized accuracy is reported when available; otherwise, standard accuracy is adopted as the evaluation metric.

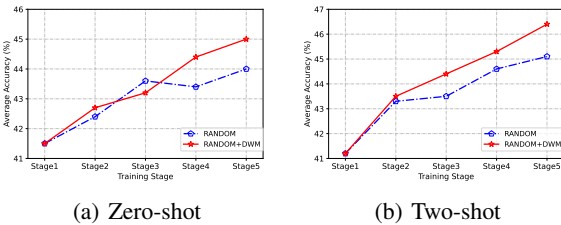

(a) Zero-shot       (b) Two-shot

*Figure 2.* Multi-stage performance of the 370M model using randomly-selected data with and without DWM.

## 5. Evaluation Results

In this section, we want to evaluate the effect of the proposed DWM. We conduct experiments to answer the following questions: 1) Is it necessary to consider the data utilization in model pre-training with data selection? 2) What is the transferring performance of the DWM to larger models or other selected datasets? 3) How does the data preference change of the trained model during training? 4) What is

---

[1] https://github.com/EleutherAI/lm-evaluation-harness

*Table 1.* Zero-shot performance of 370M pre-trained models using random-selected data with and without DWM

| STAGES | | ARC-C | ARC-E | BOOLQ | H.S. | LOQIQA | OBQA | PIQA | SCIQ | W.G. | AVERAGE |
|---|---|---|---|---|---|---|---|---|---|---|---|
| STAGE 2 | W/O | 22.0 | 39.3 | 53.2 | 33.2 | 25.2 | 28.8 | 63.8 | 65.2 | 51.1 | 42.4 |
| | W/ | 23.2 | 40.4 | 55.6 | 33.2 | 26.1 | 28.2 | 62.5 | 63.1 | 52.4 | 42.7 |
| STAGE 3 | W/O | 23.8 | 41.0 | 58.1 | 35.0 | 26.4 | 27.2 | 64.4 | 64.7 | 51.5 | 43.6 |
| | W/ | 22.5 | 41.6 | 50.3 | 34.5 | 26.3 | 30.2 | 64.4 | 66.9 | 52.3 | 43.2 |
| STAGE 4 | W/O | 24.0 | 40.7 | 52.9 | 36.1 | 25.8 | 27.6 | 64.6 | 69.7 | 49.3 | 43.4 |
| | W/ | 22.8 | 41.9 | 58.4 | 35.8 | 25.4 | 30.0 | 65.9 | 66.5 | 52.6 | 44.4 |
| STAGE 5 | W/O | 24.1 | 41.2 | 52.7 | 36.8 | 26.6 | 28.0 | 65.2 | 70.9 | 50.8 | 44.0 |
| | W | 24.3 | 42.5 | 59.9 | 36.4 | 26.4 | 29.8 | 65.3 | 68.1 | 52.7 | 45.0 |

*Table 2.* Two-shot performance of 370M pre-trained models using random-selected data with and without DWM

| STAGES | | ARC-C | ARC-E | BOOLQ | H.S. | LOQIQA | OBQA | PIQA | SCIQ | W.G. | AVERAGE |
|---|---|---|---|---|---|---|---|---|---|---|---|
| STAGE 2 | W/O | 22.9 | 41.5 | 48.3 | 33.0 | 26.6 | 27.2 | 63 | 75.9 | 50.9 | 43.3 |
| | W/ | 22.9 | 41.9 | 55.0 | 32.9 | 25.2 | 25.4 | 63.4 | 73.1 | 51.8 | 43.5 |
| STAGE 3 | W/O | 24.8 | 44.0 | 41.8 | 34.9 | 25.8 | 28.2 | 64.3 | 76.2 | 51.7 | 43.5 |
| | W/ | 23.8 | 44.4 | 49.3 | 34.9 | 24.7 | 28.4 | 63.8 | 78.3 | 52.2 | 44.4 |
| STAGE 4 | W/O | 24.1 | 45.3 | 53.7 | 35.9 | 22.3 | 28.2 | 64.6 | 76.4 | 50.8 | 44.6 |
| | W/ | 23.3 | 45.4 | 53.9 | 35.9 | 24.4 | 28.0 | 64.3 | 80.6 | 51.8 | 45.3 |
| STAGE 5 | W/O | 25.5 | 46.6 | 51.6 | 36.6 | 22.9 | 28.4 | 65.0 | 78.9 | 50.8 | 45.1 |
| | W | 24.7 | 46.8 | 56.6 | 36.5 | 25.8 | 28.2 | 65.0 | 80.5 | 53.4 | 46.4 |

affecting the effect of DWM? In the following, we will answer the questions above one by one.

## 5.1. Effectiveness of DWM

To evaluate the impact of data utilization in addition to data selection, we compare the performance of the 370M model with and without the weighting model, using randomly selected data from the SlimPajama dataset. Since we split the whole training process into 5 stages, here we provide the performance of these two approaches in the end of each stage.

Results are shown in Table 1, 2 and Figure 2, where the zero-shot and two-shot performance in the downstream tasks mentioned above are provided. Results demonstrate that the model trained with DWM (RANDOM+DWM) is better than that trained without weighting model but only randomly-selected data (RANDOM), illustrating the importance of considering data utilization. Specially, in Figure 2, we found that the weighting model generally has a more pronounced effect on model performance in the later stages of training. In contrast, during the early stages of training, the difference in performance between using and not using the weighting model is relatively minor. Additionally, the DWM not only enhances the model's performance on reading comprehension tasks, such as SciQA and BoolQ, but also improves its performance on commonsense reasoning tasks, such as the WinoGrande. Moreover, we found that compared to the model's direct generalization capability (zero-shot performance), DWM primarily enhances the model's few-shot ability. Its performance under the few-shot testing setting consistently outperforms models trained without the weighting model. We speculate that this may be because the validation set used to train the weighting model is LAMBADA,

which mainly helps improve the model's text understanding and prediction ability and is thus more suitable for such in-context learning settings.

## 5.2. Transferability of DWM

Note that in Sec. 5.1, we evaluate the effect of DWM in a 370M model and random-selected data. However, we want to emphasize that DWM is agnostic to the model size or data selection methods, operating in an orthogonal manner with respect to these factors. Hence, in the following, we aim to justify the generalization ability of the weighting model, and we transfer the weighting model trained before based on the randomly-selected data and 370M models to other data selection methods or larger models. Specially, here we apply DWM to the data selected by DSIR and QuRating, and transfer DWM to a larger Llama2-style model with 1.3B parameters. Note that the model with 1.3B parameters is commonly studied to verify the effect of different data selection methods (Wettig et al., 2024; Yu et al., 2024; Xie et al., 2023).

Firstly, we provide the zero-shot and two-shot performance of DWM when transferred to different data-selection methods, and results are shown in Table 8 and Table 3. Results demonstrate that the DWM achieves consistent performance improvements, whether applied to randomly selected data or carefully curated data like DSIR.

Moreover, to investigate the effectiveness of DWM in larger models, we transfer the weighting model trained in the 370M model to a larger model with 1.3B parameters directly. We provide the comparison in Table 9 and 4, where the zero-shot and two-shot ability of the model trained with DWM and different data-selection methods are provided. To

*Table 3.* Two-Shot performance of 370M pre-trained models using different selected data with and without DWM.

| METHOD | ARC-C | ARC-E | BOOLQ | H.S. | LOQIQA | OBQA | PIQA | SCIQ | W.G. | AVG |
|---|---|---|---|---|---|---|---|---|---|---|
| RANDOM | 25.5 | 46.6 | 51.6 | 36.6 | 22.9 | 28.4 | 65.0 | 78.9 | 50.8 | 45.1 |
| RANDOM+DWM | 24.7 | 46.8 | 56.6 | 36.5 | 25.8 | 28.2 | 65.0 | 80.5 | 53.4 | **46.4** |
| DSIR | 23.6 | 45.7 | 58.6 | 35.9 | 24.9 | 26.4 | 65.2 | 74.9 | 52.3 | 45.3 |
| DSIR+DWM | 24.9 | 46.3 | 60.0 | 36.0 | 25.8 | 29.2 | 65.3 | 78.4 | 51.5 | **46.4** |
| QURATING | 27.9 | 56.6 | 58.6 | 38.1 | 25.0 | 32.0 | 63.6 | 82.3 | 52.5 | 48.5 |
| QURATING+DWM | 28.1 | 55.6 | 59.7 | 37.7 | 24.1 | 31.2 | 63.3 | 84.6 | 53.1 | **48.6** |

*Table 4.* Two-Shot performance of 1.3B pre-trained models using different selected data with and without DWM. Unless otherwise specified, the data size is 30B tokens.

| METHOD | ARC-C | ARC-E | BOOLQ | H.S. | LOQIQA | OBQA | PIQA | SCIQ | W.G. | AVG |
|---|---|---|---|---|---|---|---|---|---|---|
| RANDOM_60B | 28.7 | 55.9 | 58.9 | 48.7 | 23.7 | 30.8 | 70.8 | 89.9 | 54.9 | 51.4 |
| RANDOM | 25.1 | 48.9 | 56.0 | 40.7 | 26.6 | 28.0 | 67.3 | 81.4 | 54.2 | 47.6 |
| RANDOM+DWM | 25.1 | 53.3 | 51.1 | 44.8 | 25.7 | 30.8 | 68.7 | 85.7 | 53.0 | **48.7** |
| DSIR | 27.7 | 53.6 | 49.7 | 44.1 | 24.6 | 31.4 | 68.8 | 85.5 | 52.8 | 48.7 |
| DSIR+DWM | 28.2 | 54.3 | 51.0 | 43.3 | 26.7 | 30.6 | 67.4 | 82.9 | 54.1 | 48.7 |
| QURATING | 33.3 | 60.8 | 61.7 | 39.3 | 25.4 | 32.6 | 61.9 | 86.9 | 50.7 | 50.3 |
| QURATING+DWM | 32.0 | 62.2 | 54.5 | 43.5 | 27.7 | 32.4 | 65.9 | 88.0 | 53.0 | **51.0** |

verify the method's improvement in model pre-training efficiency, we also provide the performance of models trained on the dataset that is twice the size (60B tokens), *i.e.*, RANDOM_60B. Firstly, results justify that DWM trained in small models can be transferred to the larger model directly, which helps improve the pre-training efficiency effectively. Moreover, we also present results for the 1.3B model trained on data selected by SOTA baselines, such as DSIR and QuRating, both with and without the DWM module. Results illustrate that the improvement of DWM when applied to different kinds of selected data is consistent across different model scales.

Specially, we found that, in 370M models shown in Tale 3, the improvement of DWM when applied to QuRating is minor, while applied to DSIR is more significant. However, in the 1.3B model shown in Table 4, the opposite trend is observed. And the marginal average improvements in these transfer settings result in mixed task-wise outcomes. We suspect this is due to the incompatibility between the model and the high-quality reasoning data during training. Here, we use data selected by QuRating along the educational dimension, which includes a significant amount of high-quality step-by-step reasoning data. For smaller models, such high-quality data can quickly lead to training saturation, leaving limited room for further performance gains through DWM. In contrast, larger models have a greater capacity to absorb high-quality data, enabling DWM to further exploit the potential advantages of such data for model training when applied on QuRating's high-quality selections. Similarly, since DSIR selects data based on its similarity to Wikipedia and books, the chosen data may contain fewer high-quality

reasoning examples. As a result, its direct application to larger models provides limited performance improvements compared to randomly selected data, leaving less room for DWM to further optimize the data selected by DSIR in the context of larger models.

It is worth noting that in DWM, a bi-level optimization strategy is employed on the 370M model to separately train the weighting model and the language model. Once the training is completed, the learned weighting model can be directly transferred to larger models without additional training. Besides, using a trained data weighting model for model training does introduce additional computational overhead. Referring to (Hoffmann et al., 2022), the training cost in FLOPs can be approximated as:

$$\text{Training FLOPs} \approx C \times \text{Model Parameters} \times \text{Token Count}, \tag{9}$$

where the constant $C$ depends on whether back propagation is performed. In our case, since the weighting model only performs forward inference when assisting the training of larger models, $C$ can be approximated as 2 (compared to 6 for full back propagation). Therefore, when transferring the 370M weighting model to the 1.3B model, the additional training overhead is roughly 9%, and this overhead continues to decrease as the size of the target model increases.

### 5.3. Analysis of Model Dynamic Data Preference

In normal large language models training, the training data are often pro-collected and treated equally important, with little attention given to selectively emphasizing or de-emphasizing specific data based on its value for improving

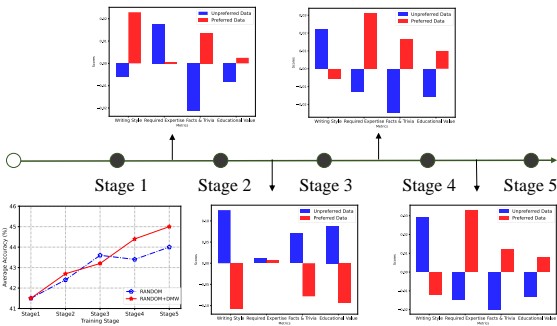

Figure 3. Preferred (red) and unpreferred (blue) data of the weighting model in different training stages, considering properties of writing, expertise, facts and educational values.

generalization, reducing bias, or enhancing specific capabilities. On the contrary, DWM aims to capture the dynamic data preference during model training, which focuses on the interaction of data as well as the shifty data preference of model during training. Therefore, in this portion, we want to analyze the change of the model's data preference, and provide more insights in the role of different kinds of data during model training.

Here we analyze the preferred and unpreferred data of DWM during different stages of training, and provide the results in Figure 3 and Figure 5. Specially, to investigate the properties of data preferred by the weighting model, we utilize the open-source scoring model from QuRating to analyze the scores of preferred (and non-preferred) data across different dimensions. Concretely, we choose a batch size of 4096 data samples, and use weighting model obtained by DWM in different training stages to weigh these data, and split the data into two sections, *preferred data* and *unpreferred data* according to their weights. Then we evaluate the prosperity of these two kinds of data, including their writing style, required expertise, facts and trivia included, as well as educational values. Results in Figure 3 demonstrate that after initial training stages like stage 1, the weighting model prefers to favor data that performs well across all these four dimensions (preferred data consistently scores positively in these dimensions), and data requiring expertise or possessing educational value is not prioritized as highly. However, counter-intuitively, during the later stages of training, data with a better writing style is less preferred and data required expertise are more preferred, which provides new insights for data selection strategies.

Specially, we also note that the weighting model after the stage 2 is inconsistent with those of the weighting model in subsequent stages, which also leads to a performance drop compared to uniformly utilizing randomly-selected data (as shown in the bottom-left corner of the Figure 3). This demonstrates that data preferences during training are

Table 5. Comparison of using different validation tasks in DWM. DWM_VAL means using the held-out validation dataset.

| SETTING | METHOD | READING | REASONING | QA | AVG |
|---|---|---|---|---|---|
| | RANDOM | 43.1 | 50.9 | 28.0 | 44.0 |
| ZERO-SHOT | R+DWM_VAL | 43.7 | 51.0 | 29.5 | 44.6 |
| | R+DWM | 44.2 | 51.5 | 29.8 | 45.0 |
| | RANDOM | 45.1 | 50.8 | 28.4 | 45.1 |
| TWO-SHOT | R+DWM_VAL | 46.8 | 50.8 | 28.2 | 46.1 |
| | R+DWM | 46.9 | 51.6 | 28.2 | 46.4 |

indeed closely tied to the model performance. And in this work, we primarily aim to highlight the importance of leveraging data during model training, as well as the effectiveness of transferring the trained weighting model. How to better obtain a weighting model that consistently improves performance throughout the entire training stages remains a topic for future research. The analysis of the preferred data of the weighting model across different data domains is provided in Figure 5 in Appendix B.2.

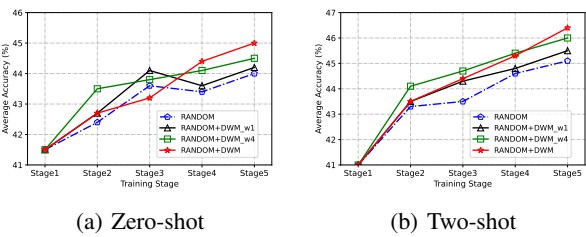

|  (a) Zero-shot  |  (b) Two-shot  |

Figure 4. Comparison of static and dynamic weighting model. DWM_w1 means only using the weighting model of the first stage. DWM_w4 means using the weighting model in the last stage.

### 5.4. Ablation Study

Note that DWM emphasizes to learn the data preference dynamically, and uses a bi-level optimization framework to help the weighting model training. In this portion, we will evaluate the importance of these two factors. Firstly, since DWM learns data preference by multi-stage alternative iterations, we compare DWM with models trained with fixed weighting model. Here we present the result of models trained with only the weighting model after the first stage (RANDOM+DWM_w1) or in the last stage (RANDOM+DWM_w4). Results are shown in Figure 4. Referring to Figure 3, in the early training stages (RANDOM+DWM_w1), the weighting model distributes its attention more evenly across different kinds of data, which limits the model's potential in later stages. In contrast, the weighting model in the last stages (RANDOM+DWM_w4) focuses more on high-quality reasoning or expertise data. While this approach significantly boosts performance in the early stages of training, its bias toward high-quality data reduces the exploration of diverse data, ultimately leading to a performance gap compared to methods that dynamically learn data preferences throughout training (RANDOM+DWM).

*Table 6.* Two-Shot performance of 370M pre-trained models with different numbers of training stages.

| METHOD | ARC-C | ARC-E | BOOLQ | H.S. | LOQIQA | OBQA | PIQA | SCIQ | W.G. | AVG |
|---|---|---|---|---|---|---|---|---|---|---|
| RANDOM | 25.5 | 46.6 | 51.6 | 36.6 | 22.9 | 28.4 | 65.0 | 78.9 | 50.8 | 45.1 |
| DWM_2_STAGES | 24.5 | 43.9 | 57.5 | 35.1 | 25.0 | 27.6 | 64.7 | 76.5 | 52.9 | 45.3 |
| DWM | 24.7 | 46.8 | 56.6 | 36.5 | 25.8 | 28.2 | 65.0 | 80.5 | 53.4 | 46.4 |
| DWM_8_STAGES | 25.5 | 46.3 | 60.1 | 36.4 | 25.2 | 28.8 | 64.9 | 77.2 | 53.7 | 46.5 |

More details are shown in Table 10 and 11 in the Appendix.

Moreover, considering that DWM uses the bi-level optimization framework and relates the weighting model's optimization to the performance in the validation set, here we justify the rationality of this objective. Here we replace the validation task LAMBADA used in DWM with the validation dataset held-out in SlimPajama (DWM_VAL) to justify the importance of the choice of the validation task. The performance of the ablation is shown in Table 5. Results demonstrate that using a validation set independent and identically distributed (i.i.d.) with the training data is less effective than using LAMBADA, a task designed to encourage text understanding and prediction. This highlights the importance of validation task selection.

In DWM, the training process is divided into five stages. To validate this design choice, we conduct ablation studies on the number of stages. Table 14 and Table 6 report the zero-shot and two-shot performance under different stage settings. Increasing the number of stages helps DWM better capture the model's dynamic data preferences, but also introduces additional training overhead for the weighting model. The results suggest that using five stages strikes a favorable balance between performance and efficiency, implying that the model's preferences remain relatively stable within each stage.

## 6. Conclusion

In this paper, we proposed a novel bi-level optimization framework with a data weighting model, designed to improve data utilization during the training of LLMs. By dynamically adjusting the weight of selected data within each batch, DWM enables more effective data usage and enhances model performance. Our experimental results demonstrate that DWM not only improves the performance of models trained with carefully selected data but also enables models trained with randomly selected data to achieve competitive results. Additionally, we show that transferring DWM to larger models yields consistent performance improvements, and we provide insights into how a model's data preferences evolve throughout training. This work opens new avenues for optimizing data selection and utilization in LLM training, providing a promising direction for more efficient and cost-effective model training.

## Acknowledgements

This work is supported in part by the National Science and Technology Major Project (Grant No.2022ZD0116403), and the Key Research and Development Program of China (Grant No.2023YFE0116400).

## Impact Statement

This paper presents research aimed at advancing the development of large language models. The proposed method is designed to facilitate data selection for large language model pre-training, thereby enhancing training efficiency and reducing computational costs. There are many potential societal consequences of our work, none which we feel must be specifically highlighted here.

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

## A. Training Details

The architecture details of the pre-training models with 370M and 1.3B parameters are presented in Table 7.

Table 7. Architecture of pre-training models with 370M and 1.3B parameters.

| Hyperparameter | 1.3B Model Value | 370M Model Value |
|---|---|---|
| Vocabulary Size | 32,000 | 32,000 |
| Hidden Size | 2048 | 1024 |
| FFN Hidden Size | 5504 | 2812 |
| Number of Layers | 24 | 24 |
| Number of Attention Heads | 16 | 8 |
| Number of KV Attention Heads | 16 | 8 |
| Maximum Context Window Length | 1024 | 1024 |
| Number of Parameters | 1,345,423,360 (1.3B) | 373,867,520 (370M) |

In the training process, a global batch size of 4 million tokens was utilized. The training was completed in approximately 7500 steps. The learning rate was set at $5 \times 10^{-4}$. The Adam optimizer was used, with the hyperparameters configured as $\beta_1 = 0.9, \beta_2 = 0.95, \epsilon = 10^{-8}$.

Table 8. Zero-Shot performance of 370M pre-trained models using selected data with and without DWM

| METHOD | ARC-C | ARC-E | BOOLQ | H.S. | LOQIQA | OBQA | PIQA | SCIQ | W.G. | AVERAGE |
|---|---|---|---|---|---|---|---|---|---|---|
| RANDOM | 24.1 | 41.2 | 52.7 | 36.8 | 26.6 | 28.0 | 65.2 | 70.9 | 50.8 | 44.0 |
| RANDOM+DWM | 24.3 | 42.5 | 59.9 | 36.4 | 26.4 | 29.8 | 65.3 | 68.1 | 52.7 | **45.0** |
| DSIR | 24.2 | 40.4 | 60.2 | 35.6 | 26.7 | 29.0 | 64.9 | 65.9 | 51.5 | 44.3 |
| DSIR+DWM | 23.2 | 41.2 | 61.9 | 35.8 | 26.6 | 29.0 | 65.1 | 67.5 | 52.0 | **44.7** |
| QURATING | 26.0 | 49.3 | 61.4 | 38.3 | 26.6 | 32.0 | 63.6 | 70.5 | 51.9 | 46.6 |
| QURATING+DWM | 26.2 | 50.8 | 57.7 | 37.5 | 27.3 | 33.6 | 63.7 | 72.7 | 52.2 | **46.9** |

## B. Additional Results

### B.1. Transferability of DWM

We provide the zero-shot performance of the trained model with different kinds of selected data and different scales of parameters. Results are shown in Table 8 and Table 9, where DWM demonstrates consistent performance improvements across different model sizes and various data selection methods.

### B.2. Analysis the Domain of Preferred Data

We provide the domain of the preferred data of the weighting model in different training stages in Figure 5, where our weighting model also exhibits a stronger focus on expertise data, such as arXiv, and reasoning-intensive code data, such as StackExchange, during the later stages of training. Due to the significant variance in data quality within domains and the relatively weak correlation between domain and data quality, a more detailed domain analysis will be left for future research.

### B.3. Ablation Study

Detailed experimental results of different ablations are provided below. Table 10 and 11 show the comparison of static and dynamic weighting model learning. Table 12 and 13 show the detailed performance of the weighting model with different validation tasks. Table 14 shows the zero-shot performance of a 370M model trained with different numbers of training stages.

*Table 9.* Zero-Shot performance of 1.3B pre-trained models using random-selected data with and without DWM

| METHOD | ARC-C | ARC-E | BOOLQ | H.S. | LOQIQA | OBQA | PIQA | SCIQ | W.G. | AVERAGE |
|---|---|---|---|---|---|---|---|---|---|---|
| RANDOM | 25.3 | 44.8 | 59.5 | 40.6 | 27.0 | 30.2 | 66.9 | 69.3 | 52.5 | 46.2 |
| RANDOM+DWM | 25.1 | 46.6 | 59.9 | 44.8 | 26.9 | 31.6 | 68.3 | 73.0 | 53.4 | **47.7** |
| DSIR | 25.9 | 47.0 | 56.3 | 44.2 | 25.8 | 30.4 | 68.4 | 70.9 | 52.5 | 46.8 |
| DSIR+DWM | 25.6 | 45.5 | 59.1 | 43.4 | 26.7 | 31.0 | 68.6 | 68.7 | 52.9 | 46.8 |
| QURATING | 30.0 | 55.6 | 61.1 | 39.4 | 25.8 | 33.8 | 62.0 | 76.3 | 50.9 | 48.3 |
| QURATING+DWM | 30.1 | 55.9 | 59.2 | 43.6 | 27.3 | 34.2 | 66.1 | 77.1 | 53.4 | **49.7** |

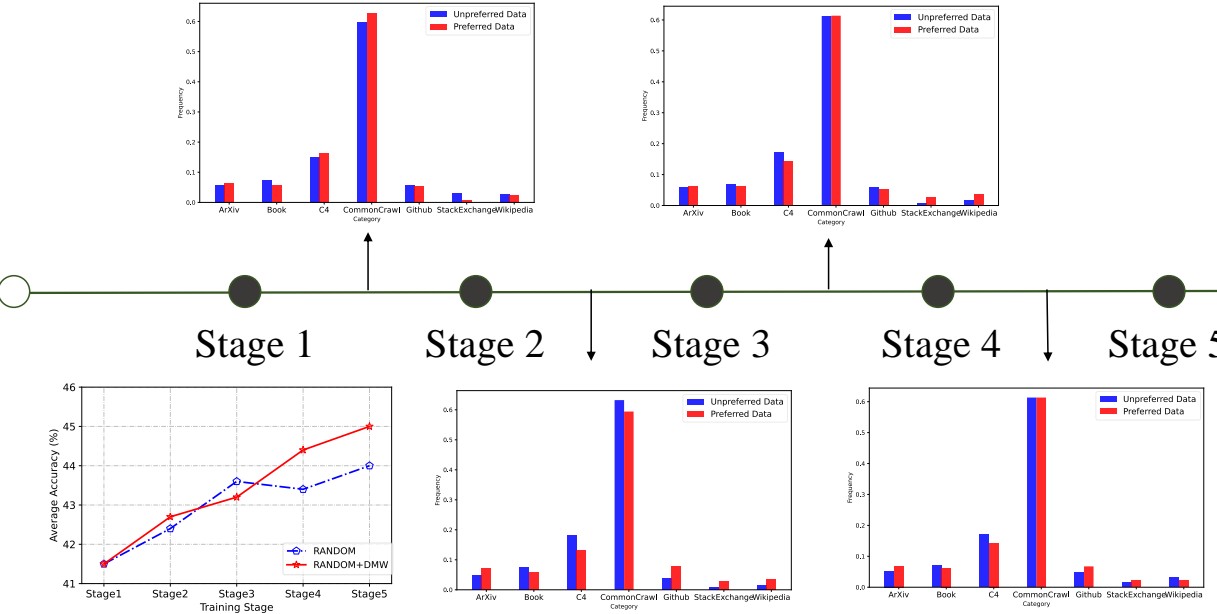

*Figure 5.* Preferred (red) and unpreferred (blue) data of the weighting model in different training stages, considering properties of different data domains.

*Table 10.* Detailed zero-shot performance of trained models using static and dynamic weighting model

| STAGE | METHOD | ARC-C | ARC-E | BOOLQ | H.S. | LOQIQA | OBQA | PIQA | SCIQ | W.G. | AVG |
|---|---|---|---|---|---|---|---|---|---|---|---|
| STAGE 2 | RANDOM | 22.0 | 39.3 | 53.2 | 33.2 | 25.2 | 28.8 | 63.8 | 65.2 | 51.1 | 42.4 |
| | R+DWM_w1 | 23.2 | 40.4 | 55.6 | 33.2 | 26.1 | 28.2 | 62.5 | 63.1 | 52.4 | 42.7 |
| | R+DWM_w4 | 23.0 | 40.2 | 61.2 | 33.1 | 26.9 | 27.8 | 64.0 | 63.3 | 51.7 | 43.5 |
| | R+DWM | 23.2 | 40.4 | 55.6 | 33.2 | 26.1 | 28.2 | 62.5 | 63.1 | 52.4 | 42.7 |
| STAGE 3 | RANDOM | 23.8 | 41.0 | 58.1 | 35.0 | 26.4 | 27.2 | 64.4 | 64.7 | 51.5 | 43.6 |
| | R+DWM_w1 | 23.0 | 41.2 | 57.2 | 35.3 | 24.6 | 30.4 | 65.0 | 66.6 | 53.5 | 44.1 |
| | R+DWM_w4 | 23.1 | 40.2 | 58.7 | 34.8 | 27.3 | 29.2 | 64.1 | 64.4 | 52.0 | 43.8 |
| | R+DWM | 22.5 | 41.6 | 50.3 | 34.5 | 26.3 | 30.2 | 64.4 | 66.9 | 52.3 | 43.2 |
| STAGE 4 | RANDOM | 24.0 | 40.7 | 52.9 | 36.1 | 25.8 | 27.6 | 64.6 | 69.7 | 49.3 | 43.4 |
| | R+DWM_w1 | 22.1 | 41.5 | 55.7 | 36.1 | 23.7 | 29.8 | 65.2 | 66.3 | 52.4 | 43.6 |
| | R+DWM_w4 | 23.1 | 40.6 | 61.5 | 35.8 | 25.5 | 29.0 | 64.6 | 64.0 | 53.1 | 44.1 |
| | R+DWM | 22.8 | 41.9 | 58.4 | 35.8 | 25.4 | 30.0 | 65.9 | 66.5 | 52.6 | 44.4 |
| STAGE 5 | RANDOM | 24.1 | 41.2 | 52.7 | 36.8 | 26.6 | 28.0 | 65.2 | 70.9 | 50.8 | 44.0 |
| | R+DWM_w1 | 22.7 | 41.5 | 56.0 | 36.8 | 25.0 | 30.6 | 65.0 | 67.7 | 52.8 | 44.2 |
| | R+DWM_w4 | 23.5 | 41.3 | 61.0 | 36.5 | 25.5 | 29.6 | 65.0 | 66.8 | 51.3 | 44.5 |
| | R+DWM | 24.3 | 42.5 | 59.9 | 36.4 | 26.4 | 29.8 | 65.3 | 68.1 | 52.7 | 45.0 |

Table 11. Detailed two-shot performance of trained models using static and dynamic weighting model

| STAGE | METHOD | ARC-C | ARC-E | BOOLQ | H.S. | LOQIQA | OBQA | PIQA | SCIQ | W.G. | AVG |
|---|---|---|---|---|---|---|---|---|---|---|---|
| STAGE 2 | RANDOM | 22.9 | 41.5 | 48.3 | 33.0 | 26.6 | 27.2 | 63.0 | 75.9 | 50.9 | 43.3 |
| | R+DWM_w1 | 22.9 | 41.9 | 55.0 | 32.9 | 25.2 | 25.4 | 63.4 | 73.1 | 51.8 | 43.5 |
| | R+DWM_w4 | 23.3 | 42.6 | 56.9 | 32.9 | 26.4 | 26.0 | 63.8 | 73.4 | 51.3 | 44.1 |
| | R+DWM | 22.9 | 41.9 | 55.0 | 32.9 | 25.2 | 25.4 | 63.4 | 73.1 | 51.8 | 43.5 |
| STAGE 3 | RANDOM | 24.8 | 44.0 | 41.8 | 34.9 | 25.8 | 28.2 | 64.3 | 76.2 | 51.7 | 43.5 |
| | R+DWM_w1 | 23.2 | 43.4 | 52.7 | 34.9 | 24.7 | 26.8 | 64.5 | 75.0 | 53.6 | 44.3 |
| | R+DWM_w4 | 23.8 | 44.0 | 49.3 | 35.2 | 27.5 | 29.4 | 64.6 | 76.9 | 51.8 | 44.7 |
| | R+DWM | 23.8 | 44.4 | 49.3 | 34.9 | 24.7 | 28.4 | 63.8 | 78.3 | 52.2 | 44.4 |
| STAGE 4 | RANDOM | 24.1 | 45.3 | 53.7 | 35.9 | 22.3 | 28.2 | 64.6 | 76.4 | 50.8 | 44.6 |
| | R+DWM_w1 | 24.2 | 44.6 | 52.4 | 36.0 | 24.1 | 28.8 | 64.9 | 76.5 | 52.1 | 44.8 |
| | R+DWM_w4 | 23.1 | 45.1 | 55.3 | 36.2 | 24.0 | 29.2 | 64.9 | 77.2 | 53.3 | 45.4 |
| | R+DWM | 23.3 | 45.4 | 53.9 | 35.9 | 24.4 | 28.0 | 64.3 | 80.6 | 51.8 | 45.3 |
| STAGE 5 | RANDOM | 25.5 | 46.6 | 51.6 | 36.6 | 22.9 | 28.4 | 65.0 | 78.9 | 50.8 | 45.1 |
| | R+DWM_w1 | 24.1 | 48.7 | 50.6 | 36.4 | 24.7 | 29.2 | 65.3 | 77.4 | 52.7 | 45.5 |
| | R+DWM_w4 | 24.1 | 46.3 | 56.5 | 36.7 | 23.4 | 29.0 | 65.6 | 80.1 | 52.6 | 46.0 |
| | R+DWM | 24.7 | 46.8 | 56.6 | 36.5 | 25.8 | 28.2 | 65.0 | 80.5 | 53.4 | 46.4 |

Table 12. Detailed zero-shot performance of trained models with DWM using different validation tasks

| METHOD | ARC-C | ARC-E | BOOLQ | H.S. | LOQIQA | OBQA | PIQA | SCIQ | W.G. | AVERAGE |
|---|---|---|---|---|---|---|---|---|---|---|
| RANDOM | 24.1 | 41.2 | 52.7 | 36.8 | 26.6 | 28.0 | 65.2 | 70.9 | 50.8 | 44.0 |
| R+DWM_VAL | 23.6 | 42.4 | 59.9 | 36.4 | 25.0 | 29.5 | 65.6 | 67.8 | 51.1 | 44.6 |
| RANDOM+DWM | 24.3 | 42.5 | 59.9 | 36.4 | 26.4 | 29.8 | 65.3 | 68.1 | 52.7 | 45.0 |

Table 13. Detailed two-shot performance of trained models with DWM using different validation tasks

| METHOD | ARC-C | ARC-E | BOOLQ | H.S. | LOQIQA | OBQA | PIQA | SCIQ | W.G. | AVERAGE |
|---|---|---|---|---|---|---|---|---|---|---|
| RANDOM | 25.5 | 46.6 | 51.6 | 36.6 | 22.9 | 28.4 | 65.0 | 78.9 | 50.8 | 45.1 |
| R+DWM_VAL | 24.8 | 46.5 | 57.0 | 36.5 | 25.3 | 28.2 | 65.1 | 80.6 | 50.8 | 46.1 |
| RANDOM+DWM | 24.7 | 46.8 | 56.6 | 36.5 | 25.8 | 28.2 | 65.0 | 80.5 | 53.4 | 46.4 |

Table 14. Zero-Shot performance of 370M pre-trained models with different numbers of training stages.

| METHOD | ARC-C | ARC-E | BOOLQ | H.S. | LOQIQA | OBQA | PIQA | SCIQ | W.G. | AVG |
|---|---|---|---|---|---|---|---|---|---|---|
| RANDOM | 24.1 | 41.2 | 52.7 | 36.8 | 26.6 | 28.0 | 65.2 | 70.9 | 50.8 | 44.0 |
| DWM_2_STAGES | 23.2 | 39.9 | 59.7 | 35.4 | 27.1 | 29.9 | 65.8 | 64.6 | 53.4 | 44.3 |
| DWM | 24.3 | 42.5 | 59.9 | 36.4 | 26.4 | 29.8 | 65.3 | 68.1 | 52.7 | 45.0 |
| DWM_8_STAGES | 25.2 | 41.7 | 60.6 | 36.0 | 27.3 | 30.6 | 65.7 | 67.3 | 52.6 | 45.2 |

