# OpenReview forum: "LLM Data Selection and Utilization via Dynamic Bi-level Optimization"
_ICML.cc/2025/Conference — ICML 2025 poster_

### Official Review · Reviewer_QCUf · 2025-03-07

**Overall Recommendation:** 4

**Summary:**

The paper proposes a dynamic bi-level optimization framework to improve data selection and utilization during LLM training. The bi-level optimization includes updating model parameters using data weighted by a weighting model, and optimizing the weighting model based on the model’s updated performance. Experiments show the proposed method enhances training efficiency and model performance, and can transfer across models and data selection methods.

## update after rebuttal

I have no more concerns, and recommend to accept this paper.

**Claims And Evidence:**

The paper claims the method improves model performance and training efficiency via dynamic data weighting. The evidence is shown in Table 1~4: the proposed DWM improve the accuracy across different model sizes and base data selection methods. Figure 3 shows DWM shifts weights from high-perplexity to diversity-focused data as training progresses.

**Essential References Not Discussed:**

No Essential References Not Discussed.

**Experimental Designs Or Analyses:**

The experiments are well organized, evaluating DWM on the diverse benchmarks, comparing against static data selection baselines (e.g, random, DSIR, MATES). The method is also transferred to larger models and other selection methods to validate generalizability. The ablation studies isolate the impact of bi-level optimization and dynamic weighting are well designed.

**Methods And Evaluation Criteria:**

Yes, the paper compares the proposed method with a series of sota compression methods. The evaluation makes sense and reasonable.

**Other Comments Or Suggestions:**

N/A

**Other Strengths And Weaknesses:**

Strengths:

-	The proposed method addresses the limitation of static data selection by dynamically adjusting data weights based on the model’s preferences.

-	The bi-level optimization framework can jointly consider effects of data samples and model update, improving data utilization effectiveness.

-	Extensive experiments demonstrates its applicability to LLM training and compatibility with other selection methods, enhancing its practical utility.

Weaknesses:

-	Bi-level optimization may introduce significant computational costs, compared to the baseline static selection methods like DSIR. Please clarify this issue.

-	The paper does not clearly show the contributions of dynamic weighting and bi-level optimization respectively, making it unclear which aspect drives the improvement.

**Questions For Authors:**

N/A

**Relation To Broader Scientific Literature:**

The work builds on static data selection methods and optimization frameworks like meta-learning, addressing their limitation of ignoring dynamic model preferences during LLM training. Deeper comparisons with adaptive sampling strategies like active learning and gradient-agnostic dynamic weighting approaches could further clarify its positioning.

**Theoretical Claims:**

No theoretical proof in the paper.

---

> ### Author Rebuttal · Authors · 2025-04-01
>
> Thank you for your thoughtful and encouraging review. Below, we address your questions and concerns in detail.
>
> ---
>
> **Q1. The Computing Cost of the Bi-Level Optimization**
>
> **A1.** DWM does introduce additional computational overhead. We provide an analysis of this issue in our response to **R3Q1** and will include the discussion in revision.
>
> ---
>
> **Q2. The discussion of the Contributions of Dynamic Weighting and Bi-Level Optimization**
>
> **A2.** Sorry for this unclear expression. We compare the contributions of dynamic weighting and bi-level optimization in Sec.5.4 and Fig. 4 in the paper. We present the result of RANDOM, as well as RANDOM_DWM_W1 and RANDOM_DWM_W4, which apply the weighting models from the first and final stages, respectively, throughout the whole training stages. Note that the weighting model used in RANDOM_DWM_W1 or RANDOM_DWM_W4 is trained by the bi-level optimization. We also include our DWM method that dynamically learns the weighting model during training.
>
> As shown in Fig. 4, although using a single-stage weighting model obtained through bi-level optimization (RANDOM_DWM_W1 or RANDOM_DWM_W4) improves model performance, dynamically learning the weighting model (DWM) allows for adaptation to the model's evolving data preferences across training stages, resulting in more robust performance gains.
>
> Thank you for your suggestion, and we will revise the writing of the paper and explicitly highlight the contributions of these two components.

---

### Official Review · Reviewer_6wW8 · 2025-03-11

**Overall Recommendation:** 4

**Summary:**

The paper introduces a Data Weighting Model (DWM) that dynamically adjusts data weights during LLM training using a bi-level optimization framework. DWM captures evolving data preferences by iteratively updating a weighting model based on validation performance. Experiments on 370M and 1.3B models demonstrate improved performance over static data selection methods, transferability across models and datasets, and insights into evolving data preferences.

**Claims And Evidence:**

Overall, both theoretical perspectives and solid empirical findings support to their claims.

**Essential References Not Discussed:**

No

**Experimental Designs Or Analyses:**

The paper presents a comprehensive experimental design. In terms of data selection, it utilizes randomly selected data and compares results with state-of-the-art data selection methods like DSIR and QuRating. For the model architecture, Llama-2 models with 370M and 1.3B parameters are employed, which are trained on 30B tokens selected from the SlimPajama dataset. The training setup uses LAMBADA as the validation set, splits training into five stages, and adopts a micro-batch size of 8. Moreover, an ablation study is conducted to compare models trained with static (fixed after early or late stages) versus dynamic weighting models, effectively highlighting the significance of continuous adaptation.

**Methods And Evaluation Criteria:**

The methods in this paper are composed of a bi-level optimization framework, dynamic data weighting, and multi-stage alternative iteration. The bi-level optimization framework jointly optimizes the LLM and the weighting model, with the weighting model updated to maximize the trained model's performance on a validation set. Dynamic data weighting is implemented by DWM, which assigns weights to data samples within each batch, considering their interactions and the model's current preferences. In the training process, the multi-stage alternative iteration alternates between updating the weighting model and the LLM parameters in stages to capture dynamic data preferences. As for the evaluation criteria, the model performance is evaluated on nine downstream tasks under zero-shot and two-shot settings, with normalized accuracy as the metric. These criteria align with standard evaluation practices in the field of data selection methods.

**Other Comments Or Suggestions:**

The costs of bi-level optimization framework should be analyzed.

**Other Strengths And Weaknesses:**

Strengths:
1.	The authors propose a novel bi-level optimization framework that effectively tackles the limitations of static data selection by explicitly modeling dynamic interactions and preferences, which is a significant advancement in the field.
2.	The paper demonstrates the transferability of DWM by achieving consistent improvements across larger models (1.3B) and a variety of data selection methods such as DSIR and QuRating, highlighting its formidable transferability and generalization ability.
3.	The authors undertake comprehensive evaluation across 9 benchmarks under zero-/few-shot settings and supplements it with ablation studies that strongly validate the design choices, ensuring the robustness and reliability of the proposed approach.
4.	This paper offers insightful analysis on how data preferences evolve during training, such as the shift from prioritizing expertise to writing quality in later stages, which adds a deeper understanding to the training dynamics.
Weaknesses:
1.	The bi-level optimization framework, while innovative, likely increases training costs. However, these additional costs are not quantified in the paper. In my opinion, the additional costs should be analyzed.

**Questions For Authors:**

N/A

**Relation To Broader Scientific Literature:**

The authors contextualize their work well within the literature on data selection, such as Qurating, DSIR.

**Theoretical Claims:**

The bi-level optimization approach effectively captures the dynamic data preferences of the model, improving data utilization and generalization. the weighting model learns to assign higher weights to data samples that are more beneficial for the model's performance, adapting as the model evolves during training.

---

> ### Author Rebuttal · Authors · 2025-04-01
>
> Thank you for your thoughtful and encouraging review. Below, we address your questions and concerns in detail.
>
> ---
>
> **Q1. The Training Cost of the Bi-Level Optimization**
>
> **A1.**  We would like to clarify that in DWM, we employ a bi-level optimization strategy on the 370M model to separately train the weighting model and the language model. Once training is completed, the learned weighting model can be directly transferred to larger models without additional training.
>
> Besides, using a trained data weighting model does introduce some computational overhead. Referring to [1], the training cost in FLOPs can be approximated as:
>
> **Training FLOPs ≈ C × Model Parameters × Token Count**,
>
> where the constant **C** depends on whether backpropagation is performed. In our case, since the weighting model only performs forward inference when assisting the training of larger models, **C** can be approximated as 2 (compared to 8 for full backpropagation). Therefore, when transferring the 370M weighting model to the 1.3B model, the additional training overhead is roughly 9%, and this overhead decreases as the size of the target model increases.
>
> Thanks, and we will add this discussion in revision.
>
> [1] Training Compute-Optimal Large Language Models, 2022.

---

> > ### Comment · Reviewer_6wW8 · 2025-04-03
> >
> > Thanks for the response. I like this work for the research area it explores. I will maintain my original rating.

---

> > > ### Author Response · Authors · 2025-04-03
> > >
> > > We sincerely appreciate the time and effort you dedicated to providing constructive feedback on our paper. Your insightful and helpful comments have offered valuable guidance for improving our work. Thanks!

---

### Official Review · Reviewer_PMBR · 2025-03-14

**Overall Recommendation:** 1

**Summary:**

This paper introduces a novel Data Weighting Model (DWM) to enhance data utilization during large language model (LLM) pre-training. DWM provides a dynamic data selection method by dynamically adjusting the weights of data samples within each training batch using a bi-level optimization framework. This framework trains a weighting model to learn the data preferences of the LLM as it evolves, allowing for more effective data utilization and improved model performance. The authors demonstrate that DWM can improves the performance of LLMs trained with randomly selected data and can also enhance existing data selection methods like DSIR and QuRating. DWM presents a promising approach for optimizing data selection and utilization.

**Claims And Evidence:**

Yes.

The experiments do support the inclusion of the method to improve performance.

**Essential References Not Discussed:**

Suchin Gururangan, Ana Marasovi´c, Swabha Swayamdipta, Kyle Lo, Iz Beltagy, Doug Downey, and Noah A Smith.
Don’t stop pretraining: Adapt language models to domains and tasks. arXiv preprint arXiv:2004.10964, 2020

Lin, Zhenghao, et al. "Rho-1: Not all tokens are what you need." arXiv preprint arXiv:2404.07965 (2024).

Liu, Qian, et al. "Regmix: Data mixture as regression for language model pre-training." arXiv preprint arXiv:2407.01492 (2024).

Yang, Yu, et al. "Smalltolarge (s2l): Scalable data selection for fine-tuning large language models by summarizing training trajectories of small models." Advances in Neural Information Processing Systems 37 (2024): 83465-83496.

Mirzasoleiman, Baharan, Jeff Bilmes, and Jure Leskovec. "Coresets for data-efficient training of machine learning models." International Conference on Machine Learning. PMLR, 2020.

Yang, Yu, Hao Kang, and Baharan Mirzasoleiman. "Towards sustainable learning: Coresets for data-efficient deep learning." International Conference on Machine Learning. PMLR, 2023.

**Experimental Designs Or Analyses:**

It seems that with addition of DWM, the performance on some tasks improve but on others decrease significantly (Table 4). It is not clear how to understand the benefit of having DWM and disadvantage.

The reliance on the LAMBADA as the sole validation set seems limiting.

**Methods And Evaluation Criteria:**

Yes.

The datasets used for evaluation and training are reasonable. The models are reasonable however small or outdated.

**Other Comments Or Suggestions:**

Figures can be more readable (Figure 2).

Time and memory complexity could be provided in the paper.

Adding more qualitative understanding (through examples) of the model's dynamic changes could be helpful to better understand the weights.

**Other Strengths And Weaknesses:**

Strengths:

- The generalizability of the method to be adapted to other selection methods is beneficial.
- The method shows some improvements in performance when adapted.

Weaknesses:

- The method requires an additional model for weighting requiring additional computation.
- Some missing literature as well as baseline methods to study (provided in the list of missing references).
- The missing baseline is missing (provided in missing literature, e.g., Data Shapley).
- The method does not provide much improvement in performance.
- Newer benchmarks could be used for evaluation of the model.

**Questions For Authors:**

It would be interesting how much additional random tokens are required to match the performance with DWM with less tokens.
Would computational time be saved?

**Relation To Broader Scientific Literature:**

The contribution of data selection is important in general and provides another method for data selection.

**Theoretical Claims:**

Not applicable.

---

> ### Author Rebuttal · Authors · 2025-04-01
>
> Review 2 Rebuttal
> ---
>
> Thank you for your thoughtful and detailed review. Below, we address your questions and concerns in detail.
>
> ---
>
> **Q1. The Missing Literature**
>
> **A1.** Thanks for your suggestions, and we have added the discussion of these related work below as well as in revision.
>
> Existing data selection methods fall into three categories: (1) **Token-level**, which filters individual tokens (e.g., Rho-1); (2) **Group-level**, which mixes data pools (e.g., Regmix); and (3) **Sample-level**, which selects individual examples. Sample-level methods include heuristic or learning-based approaches (e.g., TAPT, S2L, DSIR, QURATING) and theoretically grounded coreset-based methods (e.g., IG, CREST).
>
> Our method also belongs to the sample-level category but differs by modeling the model’s dynamic data preferences and capturing joint data effects during training. Through data weighting, DWM improves data utilization and can be easily transferred across models or combined with other selection strategies.
>
> ---
>
> **Q2. The Missing Baseline**
>
> **A2.** Thanks for your suggestion and we provided the dicussion and comparion of Data Shapley.
>
> Unlike DATA Shapley, our method (1) generalizes via end-to-end learning without per-dataset recomputation, and (2) optimizes the model directly rather than estimating fair contributions. For comparison, we trained a Shapley-based weighting model on the same data as DWM. DWM outperforms it, highlighting the benefit of learning data utility directly from the model.
>
> | method           | arc-c | arc-e | boolq | hellaswag | logiqa | obqa | piqa | sciq | winogrande | avg  |
> | :------------- | :---: | :---: | :---: | :-------: | :----: | :--: | :--: | :--: | :--------: | :--: |
> | random-shapley | 24.3  | 44.5  | 54.2  |   36.3    |  24.6  | 28.8 | 64.2 | 77.7 |    52.2    | 45.2 |
> | random-DWM     | 24.7  | 46.8  | 56.6  |   36.5    |  25.8  | 28.2 | 65.0 | 80.5 |    53.4    | 46.4 |
>
> ---
>
> **Q3. The Reliance on the LAMBADA**
>
> **A3.**  We would like to emphasize that DWM is not heavily dependent on the validation set.
>
> DWM uses LAMBADA as validation due to its common use in language model pretraining [1–2]. Other reasoning datasets, such as HellaSwag (Training set), also serve well as shown below, where DWM trained with HellaSwag validation maintains strong performance on the 370M model.
>
> | method                   | arc-c | arc-e | boolq | hellaswag | logiqa | obqa | piqa | sciq | winogrande | avg  |
> | :--------------------- | :---: | :---: | :---: | :-------: | :----: | :--: | :--: | :--: | :--------: | :--: |
> | hellaswag-training set | 24.7  | 46.8  | 56.9  |   36.5    |  26.3  | 28.2 | 64.7 | 80.9 |    51.5    | 46.3 |
> | lambada (DWM)          | 24.7  | 46.8  | 56.6  |   36.5    |  25.8  | 28.2 | 65.0 | 80.5 |    53.4    | 46.4 |
>
> ---
>
> **Q4. The Experimental Results**
>
> **A4.** Here we address the concerns related to the experimental results，which are obtained using the same model architecture or benchmark as in existing methods [2,3].
>
> **1. The Performance Improvement.** We would like to clarify that the performance gains of the DWM algorithm on average can match or even surpass the results reported in reference [1,3,4], demonstrating the effectiveness of our method.
>
> **2. The Performance Decrease of DWM in Partial Downstream Tasks**  We would like to emphasize that DWM is trained on the 370M model, and it improves the performance of the 370M model on nearly all downstream tasks. The performance drops on partial downstream tasks primarily occur when transferring the trained DWM model to specific model–data combinations, which is mainly caused by incompatibility between the training model and the training data. More detailed explaination can be found in our reply to **R1 Q3**.
>
> ---
>
> **Q5. Training Cost DWM**
>
> **A5.** DWM does introduce additional computational overhead. We provide an analysis of this issue in our response to **R3Q1** and will include the discussion in revision.
>
> In addition,  We show that DWM trained on 30B tokens matches the performance of a 370M model trained on 48B random tokens, yielding a 1.6× gain in training efficiency.
>
> | stages      | arc-c | arc-e | boolq | hellaswag | logiqa | obqa | piqa | sciq | winogrande | avg  |
> | :---------- | :---: | :---: | :---: | :-------: | :----: | :--: | :--: | :--: | :--------: | :--: |
> | random-48B  | 24.9  | 46.9  | 58.4  |   38.1    |  26.4  | 28.8 | 65.2 | 78.6 |    51.1    | 46.5 |
> | DWM         | 24.7  | 46.8  | 56.6  |   36.5    |  25.8  | 28.2 | 65.0 | 80.5 |    53.4    | 46.4 |
>
> [1] DsDm: Model-Aware Dataset Selection with Datamodels, 2024.
> [2] MATES : Model-Aware Data Selection for Efficient Pretraining with Data Influence Models, 2024.
> [3] QuRating: Selecting High-Quality Data for Training Language Models, 2024.
> [4] Data Selection for Language Models  via Importance Resampling, 2023.

---

### Official Review · Reviewer_ECYs · 2025-03-24

**Overall Recommendation:** 3

**Summary:**

This work proposes DWM, to address the limitations of existing data selection methods that ignore dynamic model training dynamics during LLM pre-training. Based on a bi-level optimization, DWM adaptively sets the weights of each data sample in a batch. In experiments, as a plug-and-play module, DWM improves the performance in downstream tasks across various settings.

**Claims And Evidence:**

1. In introduction, the authors claim the training on all available data may not be optimal and training increase the financial costs, but the authors do not conduct efficiency test on the proposed methods. On the contrary, it seems that the additionally component will even introduce much more training time, which should be properly discussed in experiments.
2. Moreover, the authors discuss the existing methods for data selection, such as selection before training and without considering the data samples in a batch indiscriminately. However, experiments show that, random sampling with DWM cannot outperform the SOTA, so I think the authors should also discuss the potential limitation of DWM.

**Essential References Not Discussed:**

N/A

**Experimental Designs Or Analyses:**

1. While in average, we observe the performance improvement, the performance of lots of tasks also decreases with DWM. Could the authors also provide some explanations which may potentially instrumental to the improvement of DWM?

2. In few-shot learning, why do authors only focus on 2 shots? What is the effect of the number of samples?

3. Moreover, the number of stages are not properly ablated.

4. Why do the authors report Table 1 and Table 2 with the 370M model? Since the authors have trained 1.3B model, reporting the performance of 1.3B model in different stages is much more convincing given that the paper focuses on LLM.

**Methods And Evaluation Criteria:**

The method design and evaluation criteria are generally convincing.

**Other Comments Or Suggestions:**

Line 261, 'dose' -> 'does'

**Other Strengths And Weaknesses:**

Strengths:
The proposed method is sound and the research topic is interesting and important.

Weaknesses:

See claim and experiments setting.

**Questions For Authors:**

If the authors are able to solve the problems in experiments and claims, I would possibly adjust my rate.

**Relation To Broader Scientific Literature:**

N/A

**Theoretical Claims:**

N/A

---

> ### Author Rebuttal · Authors · 2025-04-01
>
> Thank you for your thoughtful and valuable review. Below, we address your questions and concerns in detail.
>
> ---
>
> **Q1. Training Cost of DWM**
>
> **A1.** DWM does introduce additional computational overhead. We provide an analysis of this issue in our response to **R3Q1** and will include the discussion in revision.
>
> ---
>
> **Q2. The Potential Limitation of DWM**
>
> **A2.** As a plug-and-play module, DWM captures the model’s dynamic data preferences during training and can be applied to either randomly sampled or curated data. However, its performance is ultimately bounded by the quality of the applied data. Notably, the SOTA (QURATING) leverages data selected from 260B tokens using knowledge of GPT-3.5-turbo, offering significantly higher quality than the 30B randomly sampled tokens. As a result, DWM on random data cannot surpass QURATING, but when applied to QURATING data, it further enhances performance. We will include this discussion in the revision.
>
> **Q3. The Performance Decrease of DWM in Partial Downstream Tasks**
>
> **A3.**  First, we clarify that DWM is trained on the 370M model using randomly selected data, and it consistently improves performance across nearly all downstream tasks under both zero-shot and two-shot settings. At most, a slight performance drop occurs on a single task, a phenomenon also reported in prior work [1][2], suggesting possible optimization conflicts among downstream tasks.
>
> Second, performance degradation on certain tasks mainly arises when transferring the DWM (trained on 370M with random data) to the 370M model with QURATING data or the 1.3B model with DSIR data. This results from incompatibility between the model and the data during training.
>
> As discussed in the paper (line 354) and in reference[3], models of different scales vary in their capacity to absorb high-quality reasoning data. Furthermore, as shown in Sec. 5.3 (line 401) and Fig. 3, DWM encourages data diversity in early stages and gradually shifts toward expert and instructional data. For the 370M model with QURATING data, high-quality reasoning data quickly saturates learning, leaving little room for further improvement through DWM. Although larger models have greater capacity, DSIR data (knowledge-centric text) similarly limits DWM's optimization effect on the larger 1.3B model. Consequently, DWM yields marginal average improvements in these transfer settings, resulting in mixed task-wise outcomes. Notably, DWM performs well on the 370M model with DSIR data and the 1.3B model with QURATING data, which supports our hypothesis.
>
> These observations indicate that DWM’s effectiveness remains constrained by the applied data, highlighting the need for more robust weighting strategies adaptable to diverse data types.
>
> Thanks. We will add this discussion in revision.
>
> ---
>
> **Q4. The Two-Shot Setting**
>
> **A4.** We focus on the two-shot setting referring to exsiting data selection methods[4] to analyze the model's capacity for reasoning and generalization. In general, an appropriate number of examples can help the model better understand the task. Moreover, for models with limited capacity, an excessive number of examples may lead to overfitting on the samples and a decline in generalization ability.
>
> ---
> **Q5. The Ablation of the Number of Stages**
>
> **A5.** Thanks. We provide the ablation studies of this number below, where increasing the number of stages facilitates DWM in better capturing the model's dynamic data preferences, but it also introduces additional training overhead for the weighting model. The results show that setting the number of stages to 5 achieves a good balance between performance and efficiency, indicating that the model preferences may not dramatically change within a single stage. We will add this portion in revision.
>
> | stages            | arc-c | arc-e | boolq | hellaswag | logiqa | obqa | piqa | sciq | winogrande | avg  |
> | :--------------- | :---: | :---: | :---: | :-------: | :----: | :--: | :--: | :--: | :--------: | :--: |
> | 2-stages          | 24.5  | 43.9  | 57.5  |   35.1    |  25.0  | 27.6 | 64.7 | 76.5 |    52.9    | 45.3 |
> | 8-stages          | 25.5  | 46.3  | 60.1  |   36.4    |  25.2  | 28.8 | 64.9 | 77.2 |    53.7    | 46.5 |
>
> ---
> **Q6. Why Report the Stage-Wise Performance of 370M Model**
>
> **A6.** In our paper, we perform bi-level optimization on a 370M model, where the language model and the weighting model are trained separately. The trained weighting model is then directly transferred to a larger 1.3B model. Therefore, we report the stage-wise performance of the 370M model in Table 1 and Table 2 to analyze the effect of this bi-level optimization on model training.
>
> [1] Data Selection for Language Models  via Importance Resampling, 2023.
> [2] QuRating: Selecting High-Quality Data for Training Language Models, 2024.
> [3] Small Models Struggle to Learn from Strong Reasoners, 2025.
> [4] MATES : Model-Aware Data Selection for Efficient Pretraining with Data Influence Models, 2024.

---

### Decision · Program_Chairs · 2025-05-01

**Decision:**

Accept (poster)

**Comment:**

The paper introduces a Data Weighting Model (DWM) that dynamically adjusts data weights during LLM training using a bi-level optimization framework. DWM captures evolving data preferences by iteratively updating a weighting model based on validation performance. Experiments on 370M and 1.3B models demonstrate improved performance over static data selection methods, transferability across models and datasets, and insights into evolving data preferences.

This paper has many strengths. For example, the authors propose a novel bi-level optimization framework that effectively tackles the limitations of static data selection by explicitly modeling dynamic interactions and preferences, which is a significant advancement in the field. The paper demonstrates the transferability of DWM by achieving consistent improvements across larger models (1.3B) and a variety of data selection methods such as DSIR and QuRating, highlighting its formidable transferability and generalization ability. The authors undertake comprehensive evaluation across 9 benchmarks under zero-/few-shot settings and supplements it with ablation studies that strongly validate the design choices, ensuring the robustness and reliability of the proposed approach. This paper offers insightful analysis on how data preferences evolve during training, such as the shift from prioritizing expertise to writing quality in later stages, which adds a deeper understanding to the training dynamics.

While the reviewers had some concerns about experiments and validation set, please include the additional discussion in the next version.